# A Simple Epidemiologic Model for Predicting Impaired Neutralization of New SARS-CoV-2 Variants

**DOI:** 10.3390/vaccines11010128

**Published:** 2023-01-05

**Authors:** Giuseppe Lippi, Brandon M. Henry, Mario Plebani

**Affiliations:** 1Section of Clinical Biochemistry, School of Medicine, University of Verona, Piazzale L.A. Scuro 10, 37134 Verona, Italy; 2Clinical Laboratory, Division of Nephrology and Hypertension, Cincinnati Children’s Hospital Medical Center, Cincinnati, OH 45229, USA; 3Department of Medicine, University of Padova, 35128 Padova, Italy

**Keywords:** SARS-CoV-2, COVID-19, neutralization, antibodies

## Abstract

This study is aimed at developing a simple epidemiologic model that could help predict the impaired neutralization of new SARS-CoV-2 variants. We explored the potential association between neutralization of recent and more prevalent SARS-CoV-2 sublineages belonging to the Omicron family (i.e., BA.4/5, BA.4.6, BA.2.75.2, BQ.1.1 and XBB.1) expressed as FFRNT_50_ (>50% suppression of fluorescent foci fluorescent focus reduction neutralization test) in recipients of four doses of monovalent mRNA-based coronavirus disease 2019 (COVID-19) vaccines, with epidemiologic variables like emergence date and number of spike protein mutations of these sublineages, cumulative worldwide COVID-19 cases and cumulative number of COVID-19 vaccine doses administered worldwide at the time of SARS-CoV-2 Omicron sublineage emergence. In the univariate analysis, the FFRNT_50_ value for the different SARS-CoV-2 Omicron sublineages was significantly associated with all such variables except with the number of spike protein mutations. Such associations were confirmed in the multivariate analysis, which enabled the construction of the equation: “−0.3917 × [Emergence (date)] + 1.403 × [COVID-19 cases (million)] − 121.8 × [COVID-19 Vaccine doses (billion)] + 18,250”, predicting the FFRNT_50_ value of the five SARS-CoV-2 Omicron sublineages with 0.996 accuracy (*p* = 0.013). We have shown in this work that a simple mathematical approach, encompassing a limited number of widely available epidemiologic variables, such as emergence date of new variants and number of COVID-19 cases and vaccinations, could help identifying the emergence and surge of future lineages with major propensity to impair humoral immunity.

## 1. Introduction

Coronavirus disease 2019 (COVID-19) is a life-threatening infectious pathology that was first identified at the end of 2019 and is caused by the severe acute respiratory syndrome coronavirus 2 (SARS-CoV-2) [1]. Although the infection from this virus was associated with a considerably high risk of death upon its emergence (i.e., around 3.5%), the gradual mitigation of virus pathogenicity over time, combined with natural immunity (i.e., nearly 640 million patients have now been infected by SARS-CoV-2) and widespread COVID-19 vaccination with nearly 13 billion doses administered [2], have contributed to lowering its fatality rate to values closer to that of the seasonal flu (i.e., around 0.2%) [3]. That said, some locations had fatality rates below 0.3% before vaccination against COVID-19, whilst others maintained death rates above such value even after vaccination.

Despite the progressive trends towards lower fatality, concerns have emerged after the spread of the new and highly mutated lineages belonging to the so-called “Omicron” family, which was first officially identified (i.e., the sublineage BA.1) in November 2021 in South Africa and Botswana, though its real emergence from the former SARS-CoV-2 Delta strain is perhaps antecedent by around 2–3 months [4]. From the ancestral BA.1 sequence, multiple successive sublineages have then emerged and diverged, which have been mostly classified with the prefix “BA” followed by Arabic numbers (e.g., BA.1, BA.2, and so forth, up to BA.5), until the number of descendants (often the result of multiple recombination between preexisting Omicron sublineages) became so great that even the original classification was challenged, imposing a new formal classification encompassing additional acronyms (e.g., “BQ”, “BF”, “XBB”) [5].

The most important consequence of such an intense surge (in terms of both number and speed) of non-synonymous mutations within the SARS-CoV-2 genomic sequence (especially those encoding the spike protein) has been associated with two foremost consequences. First, the improved biological fitness of the new variants has gradually increased in terms of both better environmental resistance (i.e., lower vulnerability to high temperatures and major resistance on various surfaces) and increased affinity (and binding) to host cell receptors, thus explaining the unexpected and formerly unpredictable surge of new COVID-19 diagnoses recorded during the warmer period in 2022 in the northern hemisphere [6]. The progressive accumulation of important single-nucleotide non-synonymous substitutions within the sequence of the spike protein of SARS-CoV-2 has been associated with increasing capacity to escape from antibody-mediated neutralization, especially after the BA.4/5 sublineages became prevalent, as summarized in the article published by Chauhan et al. [7]. Specifically, the longer the phylogenetic distance from the sequence of the SARS-CoV-2 variant and/or the COVID-19 vaccine that was responsible for previous immunization, the higher the likelihood of impaired or even completely lost neutralization of a new infecting sublineage [8]. This second aspect is particularly concerning. In fact, although the risk of developing severe/critical forms of COVID-19 illness considerably declined after widespread vaccination, the immunity conferred by early immunizations with prototype monovalent vaccines may no longer be protective against the odds of (re)infection and the risk of developing severe/critical COVID-19 illness, whilst even the clinical efficacy of prophylactic or curative therapies based on administration of monoclonal antibodies may be impaired, as recently underpinned for the new BA.2.75, BQ.1 and XBB.1 Omicron sublineages [9,10,11,12].

The accurate study of disease epidemiology is essential for minimizing the impact of human pathogens on public health [13]. To this end, the use of mathematical models for estimating the epidemics of infectious diseases, for predicting the clinical course and for anticipating the clinical and social consequences of the disease itself following hospital discharge has emerged during the past decades, and has now found suitable application during the ongoing COVID-19 pandemic [14]. Besides these conventional uses, epidemiologic models may also find theoretical application for anticipating the emergence of new variants, characterized by accumulation of non-synonymous mutations that enable their escape from previous natural or vaccine-elicited immunity. To verify this hypothesis, in this proof of concept study we developed here a simple epidemiologic model that could help predict impaired neutralization of new SARS-CoV-2 variants.

## 2. Materials and Methods

Recent data on serum or plasma neutralization of the most recent SARS-CoV-2 sublineages belonging to the Omicron family, and thus encompassing (in order of emergence) the BA.4/5, BA.4.6, BA.2.75.2, BQ.1.1 and XBB.1 strains were captured from the article of Kurhade et al. [15]. Briefly, the authors assayed the neutralizing activities of human serum panels collected from 25 subjects without evidence of SARS-CoV-2 infection who received four standard doses of monovalent mRNA-based mRNA-1273 or BNT162b2 vaccines (samples drawn 23–94 days after last vaccination) and calculated the mean value, which was one of the dependent variables used in our study. The neutralization titer of these sera was assayed using a fluorescent focus reduction neutralization test (FFRNT), and neutralization was finally reported as dilution folds capable of neutralizing 50% of the different SARS-CoV-2 variants tested (i.e., FFRNT_50_). Additional information on the population and test is directly available in the original article [15].

The epidemiologic information for developing a model designed to predict impaired neutralization (i.e., FFRNT_50_ reduction) of the most prevalent SARS-CoV-2 sublineages belonging to the recent Omicron family up to present time (i.e., encompassing periods of emergence of BA.4/5, BA.4.6, BA.2.75.2, BQ.1.1 and XBB.1) was retrieved from the official statistics of the World Health Organization (WHO). More specifically, the cumulative number of COVID-19 cases and COVID-19 vaccine doses were those of the WHO coronavirus (COVID-19) dashboard [2], whilst the date of the first detection of each new highly prevalent Omicron sublineage (i.e., >10% prevalence) and the number of mutations within their spike protein compared with the original prototype sequence were identified from online resources made available by the WHO SARS-CoV-2 variants dashboard [16] and the Bacterial and Viral Bioinformatics Resource Center (BV-BRC) website [17].

The potential association between FFRNT_50_ for the different SARS-CoV-2 Omicron sublineages, date of emergence and number of spike protein mutations of the sublineages, along with the cumulative worldwide COVID-19 cases and cumulative number of COVID-19 vaccine doses administered worldwide at the time of emergence of each SARS-CoV-2 Omicron sublineage was initially assayed using univariate analysis with Pearson’s correlation. A multiple linear regression analysis was then conducted, including FFRNT_50_ as dependent variable, and all other parameters that were significantly associated with FFRNT_50_ in univariate analysis as independent variables, with final construction of a mathematical predictive model.

Statistical significance was set at *p* < 0.05. The statistical analysis was performed using Analyse-it (Analyse-it Software Ltd., Leeds, UK). The study was performed in accordance with the Declaration of Helsinki, under the terms of relevant local legislation, using publicly available repositories, such that no informed consent or Ethical Committee approvals were necessary.

## 3. Results

The data used for this our analysis, as retrieved from the official sources previously mentioned, are summarized in Table 1.

The number of mutations within the spike protein of each SARS-CoV-2 Omicron sublineage, along with the cumulative number of COVID-19 cases and COVID-19 vaccine doses administered worldwide displayed an increasing trend over time, whilst the value of FFRNT_50_ exhibited a progressive decline from the BA.4/5 sublineages that first appeared in January 2022, to the last XBB.1 strain that was first described in August 2022. In univariate analysis, the FFRNT_50_ value for the different SARS-CoV-2 Omicron sublineages was found to be significantly associated with all such variables except with the number of mutations in the spike protein, as shown in Figure 1 and Table 2.

In multivariate analysis, the FFRNT_50_ value for the different SARS-CoV-2 Omicron sublineages remained significant, and thus independently associated with all the three variables (Table 2).

Importantly, the following equation: “−0.3917 × [Emergence (date)] + 1.403 × [COVID-19 cases (million)] − 121.8 × [COVID-19 Vaccine doses (billion)] + 18,250”, predicted the FFRNT_50_ value of these five SARS-CoV-2 Omicron sublineages with 0.996 accuracy (R^2^ = 0.99, 0.98–1.00; *p* = 0.013) (Figure 2).

## 4. Discussion

Predicting the emergence of highly transmissible and/or aggressive SARS-CoV-2 variants is an essential enterprise for enabling the adoption of physical or medical countermeasures that could help containing the outbreak. Thus, the effort to construct accurate mathematical models that would help predicting the surge of new viral variants, or sublineages characterized by enhanced propensity to escape previous immunity, is pivotal for the future management of the ongoing COVID-19 pandemic [18].

This study, which was necessarily limited to the analysis of the most recent sublineages of the Omicron famil, included within the same neutralization study published by Kurhade et al. [15], demonstrated that mathematical models based on common epidemiologic variables could be developed over time that may help identifying the emergence of new sublineages or completely new variants which possess a high propensity to escape from previous immunity. This would unquestionably provide a valuable aid for tailoring future non-pharmaceutical measures (e.g., social distancing, use of physical protections such as face masks and so forth) and therapeutic interventions (e.g., development of new vaccines, updating mAb-based therapies, identifying new antiviral agents) for enabling more effective management of new outbreaks caused by highly mutated and/or divergent viral strains [19,20]. A mathematical approach to infectious disease challenges is not novel, as the use of modelling has been now advocated or even endorsed for evaluating epidemiologic behaviors during epidemic or even pandemic situations such as that currently generated by COVID-19 [21].

There are some reasonable explanations to justify the associations that we observed or not between the FFRNT_50_ value and the different variables included in our analysis (i.e., date of emergence and spike protein mutations of new sublineages, cumulative worldwide COVID-19 cases and cumulative number of COVID-19 vaccine doses administered worldwide at the time of SARS-CoV-2 Omicron sublineage emergence).

The first important aspect we found was that the decline of the FFRNT_50_ value for the different SARS-CoV-2 Omicron sublineages in recipients of four doses of monovalent mRNA-based COVID-19 vaccines progressed in accordance with the time of emergence of each of the five sublineages of the Omicron family included in our analysis. This aspect was easily predictable, since all living biological species are subjected to a constant ecologic pressure [22]. Human pathogens, including SARS-CoV-2, make no exception to this rule, since they gradually accumulate mutations that may arise and spread by chance (e.g., encompassing drift and founder effects) or through natural selection [23], and which have the ultimate scope of ameliorating their adaptation to the external habitat (e.g., improving resistance to environmental conditions) or by increasing their efficiency to colonize and reproduce within the host (e.g., developing higher affinity for human cell receptors, reducing intracellular antiviral defenses, improving replication efficiency, etc.). In keeping with this concept, the second important aspect that has emerged from our proof of concept study, is that the type rather than the amount of spike protein mutations would make the difference, as reflected by the fact that we failed to find an association between loss of neutralization and overall number of mutations within the SARS-CoV-2 spike protein of the five Omicron sublineages. Viral genomes may at least theoretically accumulate millions of different mutations, though only a few of these will consolidate and then become prominent [24]. Several lines of evidence now attest that identical mutations (especially in the sequence of the spike protein) have emerged in unrelated parallel SARS-CoV-2 lineages, most of which have developed under convergent evolution within the most critical functional domains, and thus accounting for improved virus stability (e.g., mutations within the N-terminal domain), or enhanced efficiency of SARS-CoV-2 in penetrating the host cells (i.e., mutations within the receptor-binding domain or the furin cleavage site) [25,26]. For example, the five Omicron sublineages that we analyzed in this report (i.e., BA.4/5, BA.4.6, BA.2.75.2, BQ.1.1 and XBB.1) share as many as 26 identical mutations in the sequence of the spike protein, plus three additional single-nucleotide polymorphisms in some critical positions (e.g., position V213G/E, G339D/H and F486V/S) [15], thus reflecting a clear mechanism of convergent evolution.

The strict association between virus circulation and the accumulation of escape mutations is another important aspect that needs to be discussed. It is rather clear that new mutations characterizing new viral (and fitter) sublineages are more likely to emerge and consolidate as the amount of circulating virus is sustained or even increases over time [23]. This straightforward concept is closely mirrored by the results of our study, since we also found an almost linear correlation between the declining neutralization of five different SARS-CoV-2 Omicron sublineages and the worldwide number of COVID-19 cases. The obvious consequence is that controlling the disease (e.g., by boosting worldwide vaccination) becomes of paramount importance to lower the likelihood that new variants will emerge and spread by evading human immunity.

Finally, we observed an association between impaired SARS-CoV-2 neutralization of newer Omicron sublineages and increasing number of COVID-19 vaccine doses administered worldwide. This phenomenon was predicted a few months after the beginning of the pandemic when many scientists, Elisabeth Mahase for example, proffered that the most crucial period of a viral outbreak is when a large number of people are vaccinated, as this will put the pathogen under even more sustained pressure to evolve [27]. This is mainly due to the fact that SARS-CoV-2 evolution through accumulation of escape genotypes is catalyzed by the presence of human anti-SARS-CoV-2 antibodies elicited by natural or artificial immunity [28]. Notably, the potential suitability of this proof-of concept study has been recently confirmed by another article, which used artificial neural networks for predicting recurrent mutations (rather than variants) in SARS-CoV-2 [29]. Briefly, the authors used a machine approach based on genomic rather than ecologic variables for identifying positions in SARS-CoV-2 genome where recurrent mutations are more likely to occur, ultimately achieving 0.79 sensitivity, 0.69 specificity and an overall area under the curve (AUC) of 0.80. 

We certainly acknowledge that this study has an important limitation. Being a dependent variable, FFRNT_50_ could have had a larger sample size for increasing the power of this study. Unfortunately, however, we could only use the mean FFRNT_50_ values, since the individual levels of the 25 participants enrolled in the study of Kurhade et al. [15] were unavailable. Moreover, we could only use FFRNT_50_ data for five Omicron sublineages, since these are the only subvariants reaching sufficient ecologic advantage that were tested in the original study of Kurhade et al. [15]. It is hoped that in the future, when new sublineages will likely emerge and become epidemiologically prevalent, the validity of this proof-of-concept study could be further confirmed.

## 5. Conclusions

The continued evolution and almost unremittent emergence of new dominant and phylogenetically divergent SARS-CoV-2 Omicron sublineages presents a public health dilemma (i.e., a so-called “permacrisis”) and reinforces the need to develop predictive tools that may be capable of quickly identifying the escape of new variants from previous immunity. The feasibility and reliability of predicting the emergence of recurrent SARS-CoV-2 mutations has been recently demonstrated in the work of Saldivar-Espinoza et al. [29], who developed a very complex mathematical model which was capable of anticipating nucleotide mutations and RNA reactivity with around 80% accuracy. We planned a similar approach for anticipating the emergence and spread of new variants of concerns, which does not detect divergence from a single nucleotide levels, but simplifies the entire process by including simple epidemiologic variables, such as date of emergence of new variants and number of COVID-19 cases and vaccinations, neither of which require complicated calculations or specific software. We have shown in this work that this simple mathematical approach could help identify with very good accuracy the emergence and surge of future strains escaping the current humoral immunity.

## Figures and Tables

**Figure 1 vaccines-11-00128-f001:**
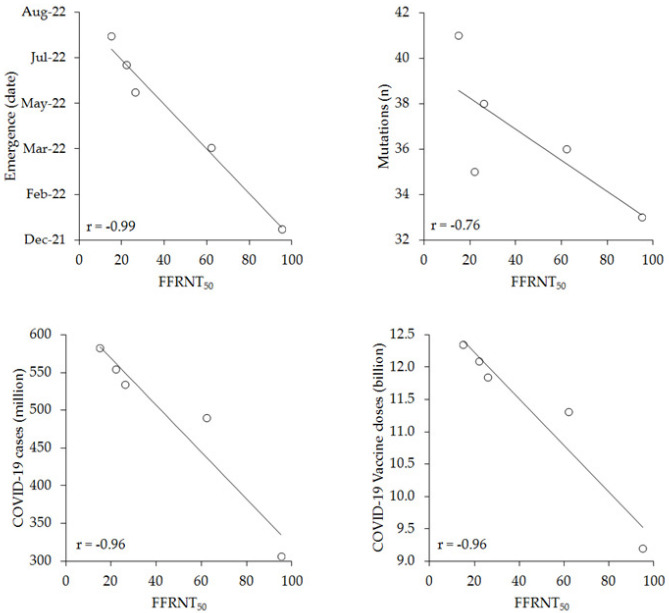
Univariate (Spearman’s) correlations between the FFRNT_50_ value for different SARS-CoV-2 Omicron sublineages in recipients of four doses of monovalent mRNA-based COVID-19 vaccines; date of emergence and number of spike protein mutations of the sublineages; cumulative worldwide COVID-19 cases; and cumulative number of COVID-19 vaccine doses administered worldwide at the time of SARS-CoV-2 Omicron sublineage emergence.

**Figure 2 vaccines-11-00128-f002:**
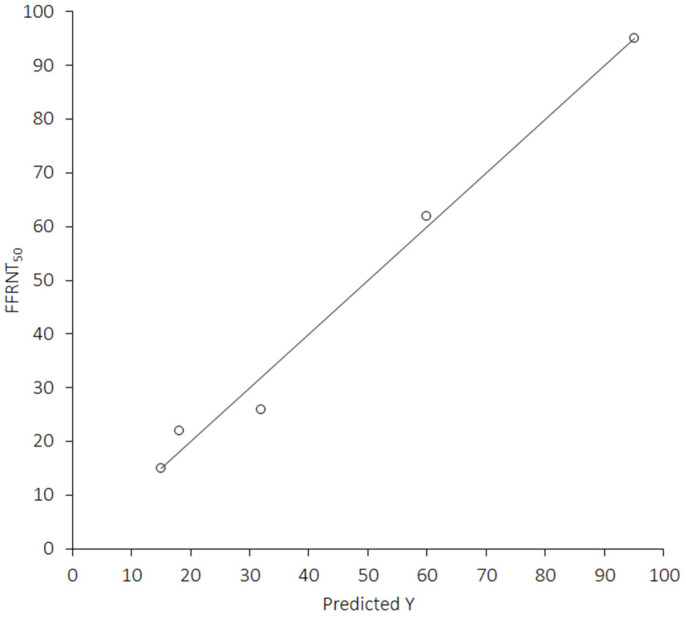
Scatter plot with fit of the mathematical model developed for predicting FFRNT_50_ value for different SARS-CoV-2 Omicron sublineages in recipients of four doses of monovalent mRNA-based COVID-19 vaccines, with incorporation of emergence date of the sublineages, cumulative worldwide COVID-19 cases and cumulative number of COVID-19 vaccine doses administered worldwide at the time of SARS-CoV-2 Omicron sublineage emergence (the full equation is reported in the text).

**Table 1 vaccines-11-00128-t001:** Data of FFRNT_50_ values for different SARS-CoV-2 Omicron sublineages in recipients of four doses of monovalent mRNA-based COVID-19 vaccines; date of emergence and number of spike protein mutations of the sublineages; cumulative worldwide COVID-19 cases; and cumulative number of COVID-19 vaccine doses administered worldwide at the time of the SARS-CoV-2 Omicron sublineage emergence.

	BA.4/5	BA.4.6	BA.2.75.2	BQ.1.1	XBB.1
Emergence (date)	22 January 2002	22 April 2002	22 June 2002	22 July 2002	22 August 2002
FFRNT_50_	95	62	26	22	15
S protein mutations (n)	33	36	38	35	41
COVID-19 cases (million)	306.957	490.213	534.040	554.251	582.439
Vaccine doses (billion)	9.20	11.31	11.85	12.10	12.35

FFRNT_50_, >50% suppression of fluorescent foci fluorescent focus reduction neutralization test; SARS-CoV-2, severe acute respiratory syndrome coronavirus 2; S protein, spike protein.

**Table 2 vaccines-11-00128-t002:** Univariate (Spearman’s) and multivariate (multiple linear) correlations between FFRNT_50_ value for different SARS-CoV-2 Omicron sublineages in recipients of four doses of monovalent mRNA-based COVID-19 vaccines; date of emergence and number of spike protein mutations of the sublineages; cumulative worldwide COVID-19 cases; and cumulative number of COVID-19 vaccine doses administered worldwide at the time of SARS-CoV-2 Omicron sublineage emergence.

	Univariate	Multivariate
Emergence (date)	−0.99 (−1.00 to −0.83); *p* = 0.001	0.003
S protein mutations (n)	−0.76 (−0.98 to 0.38); *p* = 0.139	-
COVID-19 cases (million)	−0.96 (−1.00 to −0.50); *p* = 0.001	0.004
Vaccine doses (billion)	−0.96 (−1.00 to −0.52); *p* = 0.001	0.004

FFRNT_50_, >50% suppression of fluorescent foci fluorescent focus reduction neutralization test; SARS-CoV-2, severe acute respiratory syndrome coronavirus 2; S protein, spike protein.

## Data Availability

Not applicable.

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
