# Peer review of "A Simple Epidemiologic Model for Predicting Impaired Neutralization of New SARS-CoV-2 Variants"

_vaccines, 2023, doi:10.3390/vaccines11010128_

Round 1

Reviewer 1 Report

Dear editor,

The manuscript entitled "A simple epidemiologic model for predicting impaired neutralization of new SARS-CoV-2 variants" raises an important aspect of the epidemiology of Covid-19, which is the effect of the successive mutations that have occurred in Sars-CoV-2 on population immunization. 

Major revision:

1. The authors have a good idea here of a product that can be further explored. Based on the assumption of considering the relationship between FFRNT titers and the number of mutations, number of cases of covid-19 and vaccine doses distributed as a linear and univariate relationship seems to be a very simplified way of understanding a larger phenomenon of the effect of new variants on population immunity. 

2.  In addition, an N=5 (corresponding to FFRNT50 - 4 doses of monovalent vaccines) makes the study almost unfeasible from a statistical point of view, despite pointing to a tendency towards a reduction in titers, as shown in figure 1. As a dependent variable, FFRNT50 could have a larger sample size here. The size of a sample influences the precision of your estimates and the power of the study to draw any conclusions.

3. Likewise, basing the analyzes on a single observation (one study) of the titers of 25 participants is likely to reduce the sample power and the possibilities of further conclusions.

Minor revision:

P1L41- The authors should take into account in this statement that several locations had fatality rates below 0.3% even before the introduction of vaccination against covid-19. On the other hand, some regions maintained rates above 0.3% even after the vaccine (for example, 0.4% in the city of Rio de Janeiro for 2022). 

P2L57 - what do the authors mean by "better environmental resistance"? Was the resistance of the virus to environmental conditions altered after successive mutations? Viability of the viral particle in the air or on surfaces?

P2L69-70 –isn’t the low risk of developing severe covid-19 illness associated with vaccines? even considering those developed in 2020 and 2021 before the outbreak of omicron family?

Author Response

The manuscript entitled "A simple epidemiologic model for predicting impaired neutralization of new SARS-CoV-2 variants" raises an important aspect of the epidemiology of Covid-19, which is the effect of the successive mutations that have occurred in Sars-CoV-2 on population immunization.  The authors have a good idea here of a product that can be further explored. Based on the assumption of considering the relationship between FFRNT titers and the number of mutations, number of cases of covid-19 and vaccine doses distributed as a linear and univariate relationship seems to be a very simplified way of understanding a larger phenomenon of the effect of new variants on population immunity. 

  • ANSWER: Please be aware that this is intended as a proof-of-concept study, and this aspect has been further highlighted in the paper. We have just tested the feasibility of this approach, which needs to be validated in further study (final comment added to the paper, see reply to the next point). We are otherwise thankful to the referee for the globally favourable comments on our manuscript. We’ll do our best to improve it according to the referee’s suggestions.

In addition, an N=5 (corresponding to FFRNT50 - 4 doses of monovalent vaccines) makes the study almost unfeasible from a statistical point of view, despite pointing to a tendency towards a reduction in titers, as shown in figure 1. As a dependent variable, FFRNT50 could have a larger sample size here. The size of a sample influences the precision of your estimates and the power of the study to draw any conclusions.

  • ANSWER: Reading this comment from the referee, we clearly understand that there has been a misunderstanding in what we did and this is probably due to the fact that we did not sufficiently explain our methodology, which is dependent on the data published in the original article that we used (Kurhade et al., reference 15). The samples size referring to FFRNT50 is that of the five variants included in our analysis, calculated as mean value by Kurhade et al. Therefore, it cannot be extended, because the only fiev Omicron variants that we could include in our analysis are BA.4/5, BA.4.6, BA.2.75.2, BQ.1.1 and XBB.1. We excluded BF.7, because it did never become prevalent in any country outside China and, therefore, it did not display an ecologic advantage that would enable its inclusion in our model. We have modified the text as indicated in point 3 below.

Likewise, basing the analyzes on a single observation (one study) of the titers of 25 participants is likely to reduce the sample power and the possibilities of further conclusions.

  • ANSWER: As for the explanation given in the former point, the FFRNT50 is that of the 5 variants included in our analysis, calculated as mean value by Kurhade et al., and not of the individual participants. Unfortunately, the raw value of all participants was unavailable in the article of Kurhade et al., and thereby it could not be used. To summarize, the text of the article has been changed as follows:
    • “Briefly, the authors assayed the neutralizing activities of human serum panels collected from 25 subjects without evidence of SARS-CoV-2 infection who received 4 standard doses of monovalent mRNA-based mRNA-1273 or BNT162b2 vaccines (samples drawn 23-94 days after last vaccination) and calculated the mean value, which was one of the dependent variables used in our study.”
    • “Notably, the potential suitability of this proof-of concept study has been recently confirmed by another article, which used artificial neural networks for predicting recurrent mutations (rather than variants) in SARS-CoV-2 [29]. Briefly, the authors used a machine approach based genomic rather than ecologic variables, for identifying positions in SARS-CoV-2 genome where recurrent mutation are more likely to occur, ultimately achieving 0.79 sensitivity, 0.69 specificity and an overall area under the curve (AUC) of 0.80. We certainly acknowledge that this study has an important limitation. Being a dependent variable, FFRNT50 could have had a larger sample size for increasing the power of this study. Unfortunately, however, we could only use the mean FFRNT50 values, since individuals levels of the 25 participants enrolled in the study of Kurhade et al. [15] were unavailable. Moreover, we could only use FFRNT50 data for five Omicron sublineages, since these are the only subvariants reaching sufficient ecologic advantage that were tested in the original study of Kurhade et al. [15]. It is hoped that in the future, when new sublineages will emerge and become epidemiologically prevalent, the validity of this proof-of-concept study could be further confirmed”

P1L41- The authors should take into account in this statement that several locations had fatality rates below 0.3% even before the introduction of vaccination against covid-19. On the other hand, some regions maintained rates above 0.3% even after the vaccine (for example, 0.4% in the city of Rio de Janeiro for 2022). 

  • ANSWER: Very good suggestion, thanks. Text revised accordingly, as follows: “even if some locations had fatality rates below 0.3% before the vaccination against COVID-19, whilst others maintained death rates above 0.3% even after vaccination”.

P2L57 - what do the authors mean by "better environmental resistance"? Was the resistance of the virus to environmental conditions altered after successive mutations? Viability of the viral particle in the air or on surfaces?

  • ANSWER: Good point, thanks. Text better explained, as follows: “e., lower vulnerability to high temperatures and major resistance on various surfaces”.

P2L69-70 –isn’t the low risk of developing severe covid-19 illness associated with vaccines? even considering those developed in 2020 and 2021 before the outbreak of omicron family?.

  • ANSWER: Good point, thanks. We certainly agree. Text revised, as follows: “In fact although the risk of developing severe/critical forms of COVID-19 illness considerably declined after widespread vaccination, the immunity conferred by early immunizations with prototype monovalent vaccines may no longer be protective against the odds of (re)infection and the risk of developing severe/critical COVID-19 illness”.

Reviewer 2 Report

The present study essentially consists of conducting a multiple linear regression after univariate variable selection. This procedure can be questioned. Rational selection criteria based on a presumed association would be better here than the exclusive use of a p-value. The presentation of the regression model should be more detailed (R-square).  In addition, the model found should be validated on a control set not used for modelling.

Author Response

The present study essentially consists of conducting a multiple linear regression after univariate variable selection. This procedure can be questioned. In addition, the model found should be validated on a control set not used for modelling.

  • ANSWER: Please be aware that this is intended as a proof-of-concept study, and this aspect has been further highlighted in the paper. We have just tested the feasibility of this approach, which needs to be validated in further study (final comment added to the paper). Reading this comment from the referee, we clearly understand that there has been a misunderstanding in what we did and this is probably due to the fact that we did not sufficiently explain our methodology, which is dependent on the data published in the original article that we used (Kurhade et al., reference 15). The samples size referring to FFRNT50 is that of the five variants included in our analysis, calculated as mean value by Kurhade et al. Therefore, it cannot be extended, because the only fiev Omicron variants that we could include in our analysis are BA.4/5, BA.4.6, BA.2.75.2, BQ.1.1 and XBB.1. We excluded BF.7, because it did never become prevalent in any country outside China and, therefore, it did not display an ecologic advantage that would enable its inclusion in our model. Thus, the FFRNT50 is that of the 5 variants included in our analysis, calculated as mean value by Kurhade et al., and not of the individual participants. Unfortunately, the raw value of all participants was unavailable in the article of Kurhade et al., and thereby it could not be used. To summarize, the text of the article has been changed as follows:
    • “Briefly, the authors assayed the neutralizing activities of human serum panels collected from 25 subjects without evidence of SARS-CoV-2 infection who received 4 standard doses of monovalent mRNA-based mRNA-1273 or BNT162b2 vaccines (samples drawn 23-94 days after last vaccination) and calculated the mean value, which was one of the dependent variables used in our study.”
    • “Notably, the potential suitability of this proof-of concept study has been recently confirmed by another article, which used artificial neural networks for predicting recurrent mutations (rather than variants) in SARS-CoV-2 [29]. Briefly, the authors used a machine approach based genomic rather than ecologic variables, for identifying positions in SARS-CoV-2 genome where recurrent mutation are more likely to occur, ultimately achieving 0.79 sensitivity, 0.69 specificity and an overall area under the curve (AUC) of 0.80. We certainly acknowledge that this study has an important limitation. Being a dependent variable, FFRNT50 could have had a larger sample size for increasing the power of this study. Unfortunately, however, we could only use the mean FFRNT50 values, since individuals levels of the 25 participants enrolled in the study of Kurhade et al. [15] were unavailable. Moreover, we could only use FFRNT50 data for five Omicron sublineages, since these are the only subvariants reaching sufficient ecologic advantage that were tested in the original study of Kurhade et al. [15]. It is hoped that in the future, when new sublineages will emerge and become epidemiologically prevalent, the validity of this proof-of-concept study could be further confirmed”

Rational selection criteria based on a presumed association would be better here than the exclusive use of a p-value. The presentation of the regression model should be more detailed (R-square). 

  • Good point, thanks. Table one shows the correlation value with 95%CI (we could also use the square R, if necessary, but the simple “r” already provides a valuable explanation of the correlation), whilst the square R value of the multiple regression has been included, as suggested (R2=0.99 and 95%CI, 0.98-1.00; p=0.013).

Round 2

Reviewer 1 Report

Dear editor,

After all the suggestions made during the revision of the manuscript, it is possible to perceive a visible effort by the authors to readjust the text in order to make it more understandable and clear to the readers in order to avoid misunderstandings. All points were responded and adjusted in the manuscript by the authors.

Some other questions raised during the review and which are closely related to the methodological nature of the study were duly responded by the authors. However, the authors highlighted these possible limitations in the Discussion, with similar results in more complex studies and the potential of the results produced in this paper for deeper investigations in new studies. 

Minor revision:

Figure 1 does not represent the Spearman correlation (which is represented by negative values on Table 2), but by regression lines (blue lines) and their respective errors or standard deviation (light blue). They are scatterplots between the X and Y variables.

Author Response

After all the suggestions made during the revision of the manuscript, it is possible to perceive a visible effort by the authors to readjust the text in order to make it more understandable and clear to the readers in order to avoid misunderstandings. All points were responded and adjusted in the manuscript by the authors. Some other questions raised during the review and which are closely related to the methodological nature of the study were duly responded by the authors. However, the authors highlighted these possible limitations in the Discussion, with similar results in more complex studies and the potential of the results produced in this paper for deeper investigations in new studies. 

  • We are thankful to the referee for the globally favourable comments on our manuscript.

Figure 1 does not represent the Spearman correlation (which is represented by negative values on Table 2), but by regression lines (blue lines) and their respective errors or standard deviation (light blue). They are scatterplots between the X and Y variables.

  • ANSWER: This is actually true. Sorry for exchanging the figures during manuscript preparation. The correct figures (i.e., Spearman’s’ correlation) has now been introduced.

Reviewer 2 Report

No further comments. The article can be accepted for publication in current form.

Author Response

No further comments. The article can be accepted for publication in current form.

  • We are thankful to the referee. No additional change is needed.